# Behaviour and reproduction of *Drosophila melanogaster* exposed to 3.6 GHz radio-frequency electromagnetic fields

Pieterjan De Boose[1], Felipe Oliveira Ribas[1], Duncan Bell[2], Maria Bouga[3], Eline De Borre[1], Jürg Fröhlich[4], Fani Hatjina[5], Anke Huss[6], Anastasia Kalapouti[5], Orestis L. Katsamenis[7,8], Lewis Blackwell[8,9], Pantelitsa Georgiadou[8,9], Ayesha Mohiud-din[8,9], Elham Alshamrani[8,9], Christelle Lasbleiz[10,11], Menelaos Stavrinides[12], Zoi Thanou[3], Antonios Tsagkarakis[3], Andri Varnava[12], Marco Zahner[4], Herman Wijnen[8,9], Arno Thielens[13,14]*

**1** Department of Information Technology, Ghent University, Ghent, Belgium, **2** Digital Futures Institute, University of Suffolk, Ipswich, United Kingdom, **3** Department of Crop Science, Agricultural University of Athens, Athens, Greece, **4** Fields at Work GmbH, Zurich, Switzerland, **5** Department of Apiculture, Institute of Animal Science- ELGO 'DIMITRA', Nea Moudania, Greece, **6** Institute for Risk Assessment Sciences, Utrecht University, Utrecht, The Netherlands, **7** µ-VIS X-Ray Imaging Centre, Faculty of Engineering and Physical Sciences, University of Southampton, Southampton, United Kingdom, **8** Institute for Life Sciences, University of Southampton, University Road, Southampton, United Kingdom, **9** School of Biological Sciences, University of Southampton, University Road, Southampton, United Kingdom, **10** MMDN, Univ Montpellier, EPHE, INSERM, Montpellier, France, **11** France and PSL Research University Paris, France, **12** Department of Agricultural Sciences, Biotechnology and Food Science, Cyprus University of Technology, Limassol, Cyprus, **13** Photonics Initiative, ASRC, Graduate Center of the City University of New York (CUNY), New York, New York, United States of America, **14** Department of Electrical Engineering, TU/e: Eindhoven University of Technology, Eindhoven, the Netherlands

* athielens@gc.cuny.edu

## Abstract

Insects are exposed to radio-frequency electromagnetic fields emitted by wireless telecommunication networks. A part of these fields will be absorbed by these insects. This absorption might have biological effects, depending on the amount of absorbed power. It is currently unknown at what level of absorption this might occur. To investigate this, we used RF dosimetry of adult *Drosophila melanogaster* flies, which we combined with two assays studying the locomotor activity and fecundity of *D. melanogaster* exposed to electromagnetic fields at 3.6 GHz. To perform dosimetry, we created a 3D digital twin of an adult fly using micro-CT scans of a female *D. melanogaster*. We used this model in numerical EM simulations to estimate the absorbed power in the fly as a function of RF frequency in the far field of an antenna and during the two experimental assays at 3.6 GHz. In the behavioural experiments, no effects were found on the locomotor activity for a 5-day exposure to RF field values between 5.4 and 9 V/m, which correspond to 3.56 nW to 9.88 nW absorbed power. We also did not find any effects on fecundity, at an absorption level of 1.91 mW for 48h at 3.6 GHz. In our future work, we aim to investigate possible exposure effects at higher frequencies and exposures, and for immature stages.

**Data availability statement:** The µ-CT data, the models and the XYZ used and generated for this study are accessible and can be accessed through Zenodo.org at https://doi.org/10.5281/zenodo.14838021. Researchers interested in accessing the data can use this DOI to locate and download the relevant information. Other data is readily available in the paper.

**Funding:** European Union's Horizon Europe research and innovation funding program under grant agreement No 101057216 (ETAIN).

**Competing interests:** The authors have declared that no competing interests exist.

## Introduction

Telecommunication networks facilitate data exchange between users and the network through radio-frequency electromagnetic fields (RF-EMFs). Traditionally, these transmissions have occurred across a range of frequencies, predominantly between 100 MHz and 6 GHz [1]. As these networks continue to evolve, new frequency bands are integrated, including those within the 5G New Radio (NR) spectrum, such as the 3.4–3.7 GHz band [2]. Since these fields are emitted into the environment, animals in general, and flying insects in particular, are exposed to them. In some frequency bands of 5G networks, corresponding wavelengths are similar in size to insects, which raises the concern of increased RF-EMF absorption due to possible resonance effects. In previous studies on invertebrates, RF-EMF exposure has been shown to induce behavioural changes [3] and changes in reproductive capacity [4] and development [5]. Given that insect populations and their biodiversity are already under significant pressure, it is crucial to anticipate and assess additional potential aggravating factors that may further exacerbate their decline. As insects play a vital role in pollination and ecosystem stability, this study contributes to the broader responsibility of preserving planetary health.

*Drosophila melanogaster* is an excellent model organism for assessing the impact of external physical agents such as RF-EMFs due to the large amount of genetic information available [6] in conjunction with the developmental cycle [7] and behaviour [8] being very well understood. Therefore, we decided to investigate RF-EMF exposure in this model. Our goals were to establish a relationship between exposure to RF-EMFs and absorbed power within the insect, and to assess whether this absorption is associated with changes in two key biological parameters—behavior and fecundity—both of which are responsive to environmental and physiological stressors.

In order to allow for more consistent comparisons across studies, electromagnetic dosimetry provides a more replicable and biophysically relevant measure than a description of the exposure configuration. RF-EMF dosimetry quantifies the absorption of RF-EMFs inside organisms and is usually carried out with numerical electromagnetic simulations. Using this method, it has been shown previously that absorption of RF-EMFs into insects depends on the frequency [9,10], the developmental stage of the insect [11], the size of the insect [10,12], and that insects, that fly by close to the mobile phone base stations, can absorb relatively large fractions of the power (up to 25%) emitted by such antennas [13].

The accuracy of numerical simulations depends on the quality of the digital insect model. Therefore, prior studies in this field have relied on micro-computerized tomography (CT) to obtain anatomically accurate digital insect models [9–11], which can then be combined with insect-specific dielectric properties [9] to calculate RF-EMFs inside and around an insect. To the best of our knowledge, this is the first study conducting numerical electromagnetic simulation studies for *Drosophila melanogaster*, to quantify what RF-EMF absorption these flies experience in the environment and during experimental trials. There have been prior studies on RF-EMF exposure of *D. melanogaster*, an overview of these

studies can be found in [11]. There are a series of studies that investigate the fecundity of *Drosophila* exposed to relatively low, presumably non-thermal levels of RF-EMFs between 0.1 and 6 GHz [4,14–20]. Related to this, there are a series of studies that investigate effects on the ovaries of female *Drosophila* under RF-EMF exposure in the same frequency range [19,21]. However, the studies listed above either did not use sham controls or, when they did use sham controls, they provided unreliable exposure measurements or no exposure measurement at all, making these results difficult to interpret. The lack of either dosimetry or sham control in these studies renders any effects of RF-EMF exposure on the fecundity of *D. melanogaster* inconclusive or impossible to associate with a level of RF absorption. Therefore, we will investigate this further in this paper.

Besides investigating fecundity endpoints, we also wanted to examine the impact of RF-EMF exposure on the rhythmic locomotor behavior of adult *D. melanogaster*. Analyzing locomotor behavior under constant conditions allows us to detect changes in both activity levels and their rhythmic organization, which is governed by the flies' innate circadian clock. Circadian clocks synchronize physiology and behavior to environmental time cues—such as the daily fluctuations in temperature and sunlight resulting from Earth's rotation [22]. The fact that changes in temperature act as entrainment cues for circadian rhythms [23–25] is of particular relevance. Since RF-EMF exposure in insects is known to induce dielectric heating, daily patterns of RF-EMF exposure could serve as entrainment cues for circadian locomotor behavior. Under constant conditions, where external cues are absent, any RF-EMF-induced entrainment would be readily observable, allowing us to determine whether RF-EMF exposure acts as a zeitgeber. Furthermore, while autonomous circadian rhythms in locomotor activity persist with a period close to 24 hours, most *Drosophila* lab strains display a slightly shorter free-running period in the absence of external cues [26], leading to asynchrony with the 24-hour cycle. This assay thus provides a means to assess how RF-EMF exposure might alter period length and disrupt the endogenous circadian rhythm.

Given the above, we performed two RF-EMF exposure experiments at 3.6 GHz, investigating the two potential effects of this exposure discussed above: fecundity of RF-EMF-exposed *Drosophila melanogaster*, as it is the most prevalently studied endpoint in literature, and disruption of circadian rhythm, given the potential thermal pathway and suitability of *D. melanogaster* for such experiments. We combined these with numerical simulations of the same exposure to link a potential effect to absorbed power levels in the flies.

## Methods

This section describes the methods used for the RF exposure experiments and the numerical dosimetry. The methods for the experimental, in-situ RF exposure assessment are described in the Extended Methods section of the S1 File.

### Electromagnetic simulations

There are no experimental methods available for RF dosimetry of insects. Therefore, we chose to work with numerical dosimetry in combination with experimental exposure assessment in this study. This section outlines the different steps that are taken to execute RF EM simulations of a *Drosophila* fly under RF-EMF exposure.

**Micro-CT scanning of a *drosophila melanogaster* fly.** One- to two-day-old wild-type Berlin-K adult females were fixed and stained according to [27]. Following 5 min incubation in 0.5% Triton X-100 PBS, the flies were fixed in Bouin's solution for 24h, washed in 1x PBS, and then stained for 2 days in Lugol's solution prior to washing in 1x PBS and scanning in ultrapure water. The fly was then scanned at the µ-VIS X-ray Imaging Centre of the University of Southampton (https://muvis.org). The specimen was placed in a shielded pipette tip in the µ-CT scanner and imaged in liquid in a custom Nikon XTH225ST µ-CT system, optimised for histology applications. The scan acquisition involved collecting 1001 projections, with an angular step of 0.36° between projections, some of which are shown in Fig 1.

**Drosophila model creation.** The image stack was processed in Python using dedicated libraries such as trimesh and open3d. The marching cubes algorithm [28] was employed to create 3D models in STL format. This involves an interpolation of the original images into a 3D mesh of interconnected points that make up the digital model. The model was further refined in Blender 3D software (Blender Foundation, Amsterdam, NL) to enhance mesh quality and

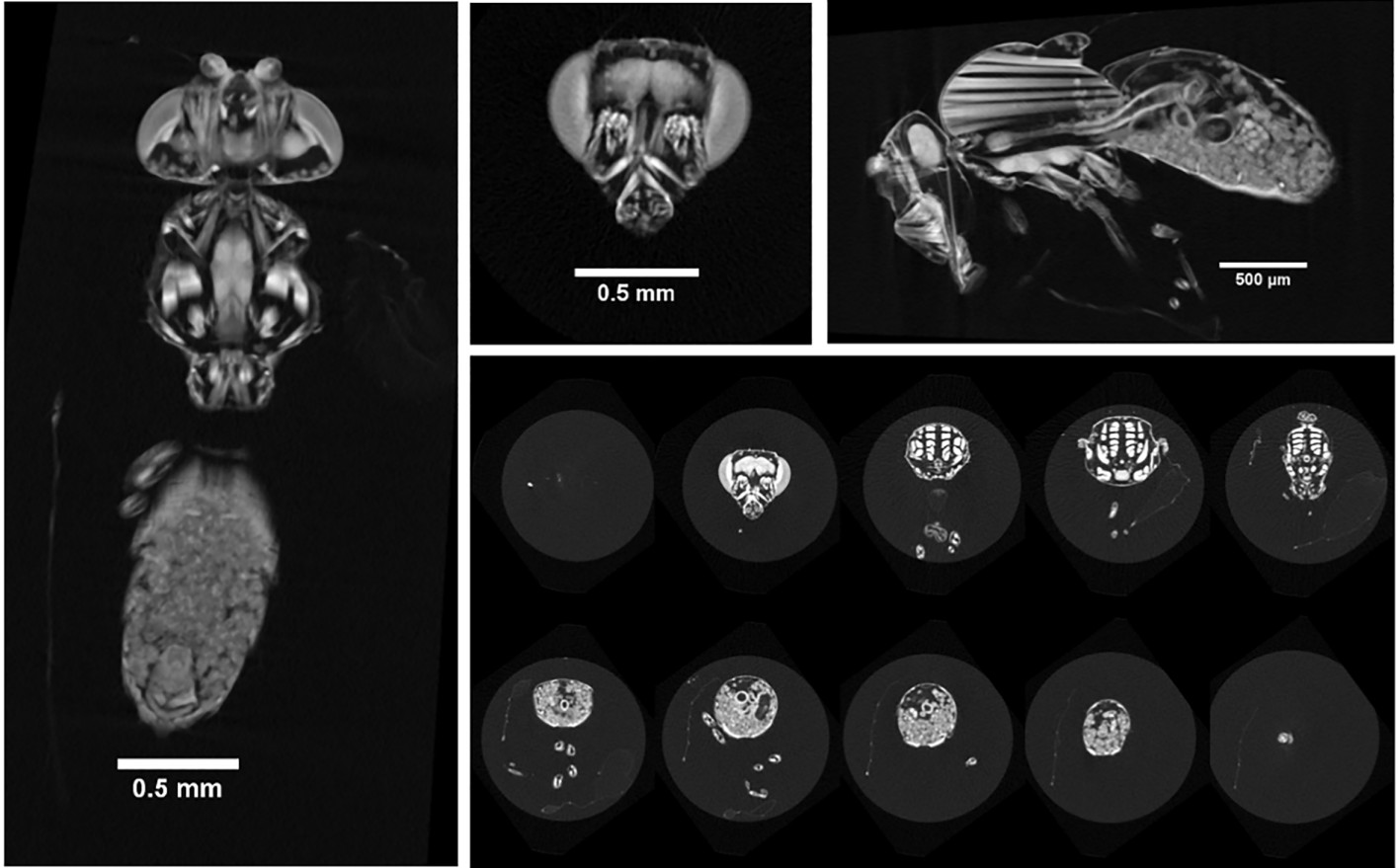

**Fig 1. 20×slice average intensity projection visualized across coronal, transverse, and sagittal planes, accompanied by a montage of transverse plane slices sampled at every 115th slice. The complete dataset is available at** https://doi.org/10.5281/zenodo.14838021.

positioning. Any holes and artefacts in the mesh were closed and the mesh was aligned with the grid used in numerical simulations. The model required repositioning, as deceased flies naturally assume a posture with closed wings and curled legs. Therefore, an armature was defined within the model, allowing for controlled repositioning of the legs and wings. Finally, different tissues were volumetrically defined within the model based on anatomical knowledge of *Drosophila melanogaster*. The following tissues were discerned and segmented (see Fig 2 (b)): inner tissue, exoskeleton, brain, halteres, wings, gonads, and muscles.

**Far-Field numerical simulations.** Finite-difference time-domain (FDTD) simulations were carried out in Sim4Life (ZMT, Zürich, Switzerland) to calculate the electromagnetic field distribution inside the D*rosophila* model and assess the electromagnetic power absorbed ($P_{abs}$) by the insect at the following frequencies: 1 GHz, 2.45 GHz, 3.6 GHz, 6 GHz, 12 GHz, 24 GHz, 60 GHz, 90 GHz, 120 GHz, and 240 GHz, when it is exposed to an antenna that is electromagnetically far away from the insect. A set of twelve plane waves incident along the *Drosophila* model's main axes with two orthogonal polarizations was used to emulate far-field exposure, see Fig 2 (a), following [9–11]. These simulations serve two purposes: they can be used to quantify the environmental exposure of a fly, and they can serve as a good proxy for the exposure experienced during the behavioral experiments.

In FDTD simulations, the *Drosophila* model and its environment are discretized along a Yee lattice. The insect model was finely meshed with a voxel size of 20 μm, ensuring compliance with the Courant stability criterion (grid step $< \frac{\lambda}{10\sqrt{\varepsilon}}$)

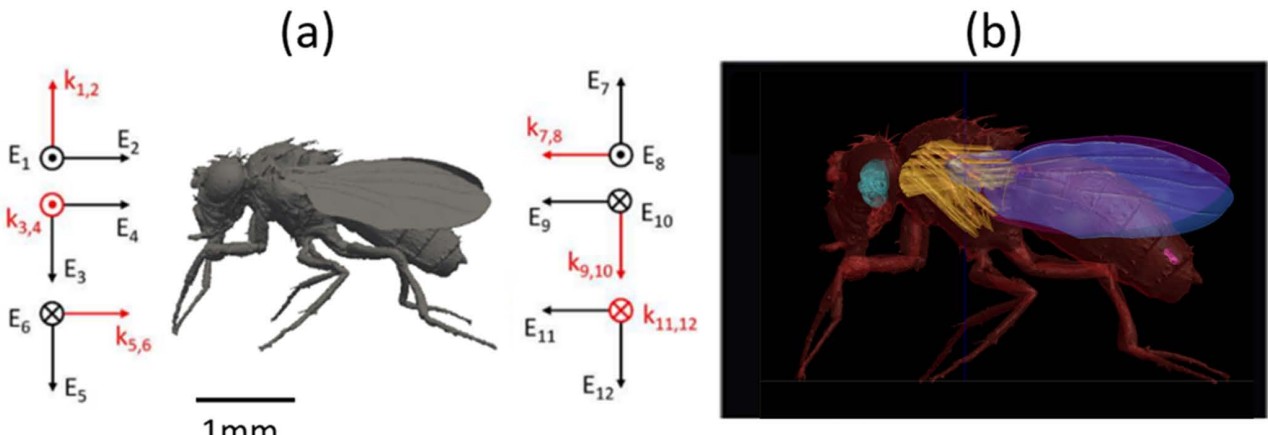

**Fig 2.** (a) The twelve polarizations considered in the far-field simulations, where k represents the wave vector and E the electric field, shown relative to a side view of the homogeneous Drosophila digital twin. (b) The heterogeneous Drosophila digital twin (right) segmented into 7 tissues: inner tissue, exoskeleton, brain, halteres, wings, gonads and muscles.

at all frequencies [29] and adhering to the Nyquist criterion by including at least three grid steps in the smallest layer to minimize numerical dispersion and enhance computational accuracy [30]. The dielectric properties of insect tissue at 3.6 GHz were determined by a Cole-Cole fit to literature data [31,32] and a recent set of coaxial probe measurements [33]. In that way, at 3.6 GHz, the permittivity was set to 42 and the conductivity to 2.8 S/m. At the other frequencies, we used the interpolated data of Thielens et al. [9]. The dielectric properties for each selected frequency are listed in S1 Table.

For the simulation to reach a steady state, the number of periods was set at 16 for frequencies up to 24 GHz, 23 for 60 GHz, 27 for 90 GHz, 33 for 120 GHz and 38 for 240 GHz. The root mean square (RMS) value for the incident electric field strength was always 1 V/m. After the simulations, the electric field distribution is extracted for each plane wave and for each frequency. The absorbed power $P_{abs}$ is then calculated as:

$$P_{abs} = \int_V \sigma \left|\overline{E}\right|^2 dV$$

(1)

Where $\sigma$ is the conductivity of the fly in (S/m) and V is the volume of the (part of) the model over which the integral is calculated. In this new model, we were able to define different tissues. Hence, each sub-integral over the tissues' volume provides the absorbed power in each tissue. The integral over the entire volume of the model provides the whole-body absorbed power. These values were calculated at each frequency. At 3.6 GHz, the absorbed power was rescaled to the exposure values in V/m that the insects experienced during the experiments in order to quantify the power absorbed during the experiments, as described in the following subsections.

**Dosimetry of behavioural experiments.** The behavioural experiments took place entirely in the far field of an emitting antenna within an incubator (see Fig 3 (a) and (b)). The absorbed power values obtained using FDTD simulations and Eq. 1 at 1 V/m, were rescaled to the measured electric field strength to estimate the absorbed power in the flies during the behavioural experiments.

**Dosimetry of fecundity experiments.** The dosimetry of this experiment involved placing the fly model in different locations in the near field of an antenna (see Fig 3 (c) and S1 Fig) resonating at 3.6 GHz. This allowed us to establish the power absorbed by the flies during these experiments. More details can be found in the Extended Methods section of the Supporting Information

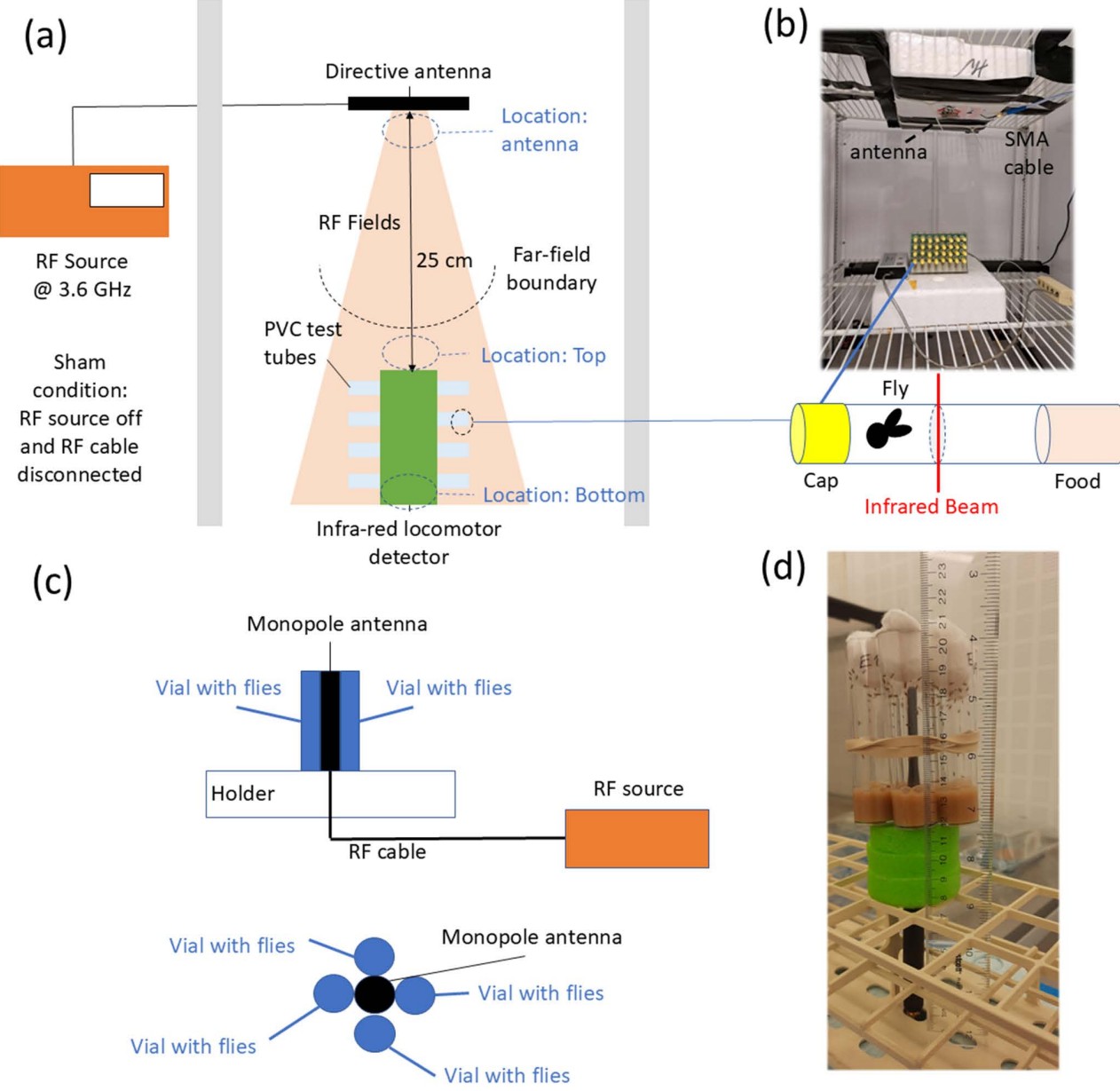

**Fig 3.** (a) Illustration and (b) image of the RF-EMF exposure setup used for behavioural experiments. (c) Illustration and (d) image of the RF-EMF exposure setup used for the fecundity experiments.

## RF exposure experiments

**Behavioural experiments.** Trikinetics DAM Drosophila activity monitors were used to assess the impact of cycles of RF exposure on daily locomotor activity levels as well as features of circadian locomotor behavior. Adult *Drosophila melanogaster* locomotor behavior was examined under constant conditions (20°C in the dark) during 5 cycles of 12h on/12h off RF exposure or sham exposure, preceded and followed by 5 days of control conditions. The locomotor behaviour patterns were examined for impact on activity levels and rhythmic features, including relative rhythmic power

and period length. Fig 3 (b) illustrates the experimental setup used. DAM monitors (Trikinetics, Waltham, MA, USA) were loaded with 32 individual fruit flies, each in a glass tube containing sugar-agar-tegosept (5%−1%−0.07% w/v) diet on one end and a cotton plug at the other [23]. Two infrared beams were projected across the midpoint of each tube, allowing timed activity events to be recorded every time a fly crossed the beams [34].

The DAM monitor was placed inside a Percival DR-36VL climate chamber (Clf, Plant Climatics, Wertingen, Germany) operated at 20°C and 70% relative humidity in constant dark conditions. This metallic chamber provided shielding from external RF-EMF exposure (see RF-EMF measurements). A directive RF-EMF-emitting antenna (Taoglass Maximus Flexible Ultra-Wide-Band Antenna, 700–6000 MHz) was positioned 25 cm above the DAM device, exceeding the far-field limit at 3.6 GHz, and suspended 6.4 cm under a metal grid. This antenna either transmitted at 15dBm and 3.6 GHz, corresponding to the 5G-NR frequency deployed in the EU and UK, or remained connected to a dummy load during non-emitting, control conditions. The antenna was powered by an RF signal generator (Aaronia Spectran V5) located outside the chamber and connected via a coaxial Sub-Miniature A-type (SMA) cable. To ensure proper separation from the metallic grid supporting it, the antenna was mounted on a Styrofoam structure. Prior to conducting experiments, exposure levels in this setup were quantified using an RF field probe.

Male and female adult Berlin-K *D. melanogaster* were loaded into the vials, and they remained there for 5 days whilst being monitored without any RF-EMF emissions. After 5 days, the second 5-day interval started in which experimental flies were exposed to 12h on (from 09:00–21:00)/12h off RF-EMF (continuous exposure) irradiation cycles. During this period, control flies remained unexposed. The experiment then concluded with 5 days of no RF-EMF exposure for all groups. The 5-5-5 days design was used as a compromise between maintaining good survival throughout the experiment and having a sufficient number of days available for each segment to determine reproducible periodicity. Sexual dimorphism of adult *D. melanogaster* locomotor activity has been reported for both light/dark and constant dark conditions with male flies exhibiting reduced mid-day activity in the former and a shortened period length in the latter conditions [35]. Therefore, male and female flies were analyzed separately. Locomotor activity data was collected using DAM system software and analyzed using CLOCKLAB (Actimetrics, Lafayette, IN, USA) with a 30-min resolution [36]. Chi-square periodogram analyses were used to determine the period length and relative rhythmic power (the ratio between amplitude and significance threshold at the detected circadian period) for each fly during each 5-day interval. The data was analyzed using GraphPad Prism (version 10.4.1, Dotmatics, Bishop's Stortford, UK). Normality was assessed using the Shapiro-Wilk test. If normality was met, repeated measures ANOVA was performed without assuming sphericity, followed by Tukey's multiple comparisons test. If normality was not met, Kruskal-Wallis tests were used with Dunn's multiple comparisons test. Kruskal-Wallis tests were conducted for all period length data and control RRP data, while repeated measures ANOVA was applied to activity count data, except for female-exposed flies. Since no significant pairwise differences in period length were observed and the 15-day average period length remained 22.5 hours across all sex-treatment combinations, average activity levels were calculated per 22.5-hour circadian cycle. Activity counts per fly were then stratified by treatment interval.

We did not perform a positive control for the behavioral experiments because it has been experimentally established that the activity of D. *melanogaster* can change over time and that commercially available DAM can detect this change [37]. We also did not include a negative control, as there is no known mechanism by which the presence of a non-emitting antenna could interact with the activity of *D. melanogaster* flies.

The flies were blind to the RF exposure as they had no way of knowing whether the RF source outside of the chamber was on or off. The data activity recording and analysis are performed automatically by the DAM device and software, which are insensitive to RF exposure conditions. Therefore, the researcher who received the data and analysis was not blind to the exposure but could also not influence the results.

**Fecundity experiments.** A second experiment investigated the impact of RF-EMF on the fecundity of *D. melanogaster*. In this experiment, 10 male and 10 female flies were placed in each of six test tubes arranged in a

concentric circle around a monopole antenna (Linx Technologies, ANT-5GMWP1-SMA). The antenna transmitted a 15 dBm RF signal at 3.6 GHz under exposed conditions. The setup was placed in a climate-controlled room, as shown in Figs 3 (b) and (c), in a 12-hour light:12-hour dark environment illuminated by fluorescent tube lights. Control flies (negative) were kept in a separate room under identical temperature and lighting conditions for the first two days. After 48 hours of continuous RF-EMF exposure or negative control, flies from each vial were transferred to a fresh vial. This transfer was repeated every 24 hours for five cycles. Humidity was controlled from day 2 onwards by co-housing experimental and control vials in the same humidified box. The number of eggs deposited was recorded after each transfer. Pupae counts were recorded two weeks after seeding, and their hatching was tracked for an additional week. Given the minimal generation time (±14 days), observations remained well within the first generation. Working with 48h of exposure allowed for the analysis of both immediate and indirect potential effects of the exposure. Fecundity, defined as the number of adult offspring per female per day, was calculated for each vial. Shapiro-Wilk tests confirmed normality. A two-way ANOVA with post-hoc tests (Tukey's multiple comparison test) was performed to assess the effects of treatment (5G versus sham), time, and their interaction on fecundity. We did not perform a positive control for the fecundity experiments because the method used here has already been experimentally established in [38]. The flies in the exposure experiment could see the antenna but could not see the RF source and thus could not know whether the antenna was emitting RF fields or not. The vials were coded, and eggs were counted without immediate knowledge of the associated treatment. Decoding was performed during statistical analyses by the same investigator who had coded the data. The investigator cannot influence the number of eggs produced by the flies.

**In-Situ RF-EMF exposure assessment.** To quantify the RF-EMF exposure during the behavioral experiments, we performed a series of in-situ electric field strength measurements with a Narda NBM Probe (Narda, Hauppage, NY) inside the incubator for both exposed and sham conditions. These measurements were repeated with the ExpoM-RF 3 (Fields at Work, Zürich, Switzerland) to obtain frequency-specific information. Mann-Whitney U tests were used to evaluate our hypothesis that most of the E-field strength inside the incubator was due to 3.6 GHz emissions and that the metal incubator effectively shields against environmental RF-EMFs.

For the fecundity experiments, the received power at 4 distances from the stub antenna was measured with the RF Explorer Probe (Seeed Studio), both with and without vials, and both for emitting and non-emitting conditions. The extended method of all these measurements, as well as their results, can be found in the Supporting Information.

## Results and discussion

### 3D Digital *Drosophila* model

Fig 1 shows cross-sections of the adult *Drosophila* fly. Good contrast with anatomical detail for most soft tissues was achieved with the Lugol's KI/I2 Iodine staining. Fig 2 shows the resulting 3D model obtained after postprocessing in Python and Blender. The model provides one with anatomical accuracy, as it is based on μ-CT scans and flexibility in terms of posture, with its flexible armature. It can be found in the Sketchfab and Zotero databases [39]. The same databases contain other whole-body models of fruit flies, but these are mainly created using photogrammetry (and contain no information on internal anatomy) or stem from artistic digital drawings of fruit flies, which do not necessarily guarantee anatomical accuracy.

There have been prior studies that used μ-CT scanning on *D. melanogaster* samples. Chen et al. [40] scanned several flies without contrast at 800 nm resolution to map out Ca-deposition in the flies. Mattei et al. [41] performed μ-CT with a resolution down to 400 nm on a series of iodine-stained 3-to-5-days-old *D. melanogaster* flies with the aim of imaging reproductive organs. They used their scans to create 3D models of the uteri of female flies. Schoborg et al [42] explored different staining methods to scan *D. melanogaster* with a resolution down to 5.5 micrometers, in mature and immature stages. While several of these studies provide higher resolutions in their μ-CT scans, none of them present a 3D model of the fly, such as the one presented in this work. When comparing the insect model created in this paper with prior work

on 3D models for electromagnetic simulations of insects [9–11,43], this heterogeneous model provides more anatomical information than the homogeneous models used in our prior studies. Even though, we currently do not possess tissue-specific dielectric properties for each of these different tissues. The mere fact that we can label internal electromagnetic fields as being induced in a specific tissue opens new avenues of RF-EMF research.

### RF-EMF dosimetry

The results of the experimental RF exposure assessment can be found in the Extended Results section of the S2 File and in the S2 and S3 Tables of the Supplementary Materials.

**Absorbed power in *D. melanogaster* under far-field RF-EMF exposure.** Fig 4 shows the whole-body averaged absorbed power as a function of frequency in *D. melanogaster*. Below 60 GHz, the absorbed power is clustered into three groups depending on the orientation of the electric field. As previously observed in other studies [12,44] the absorbed power is maximal for orientations in which the electric field is polarized along the longest axis of the model. This effect is less pronounced at frequencies above 90 GHz, as absorption values for different polarizations get relatively close to each other. At these frequencies, the theoretical penetration depth in the model decreases well below the smallest diameters of the model, which causes the internal electric field distributions in Fig 5 to become increasingly polarization-independent at high frequency.

The *Drosophila* model with a total volume of $2.1\,mm^3$ is smaller than all insects studied in [9–11] and consequently has a lower whole-body averaged absorbed power than those larger insects. The insects studied in [9,11] showed maximal absorptions between $3 \cdot 10^{-8}$ W and $2 \cdot 10^{-5}$ W for an incident field strength of 1 V/m but have volumes between $3\,mm^3$ and $1200\,mm^3$. Additionally, the larger insects showed an absorption peak at lower frequencies. Herssens et al. [12] list curves for a volumetric prediction of the far-field absorption of insects. Based on a volume of $2.1\,mm^3$, the

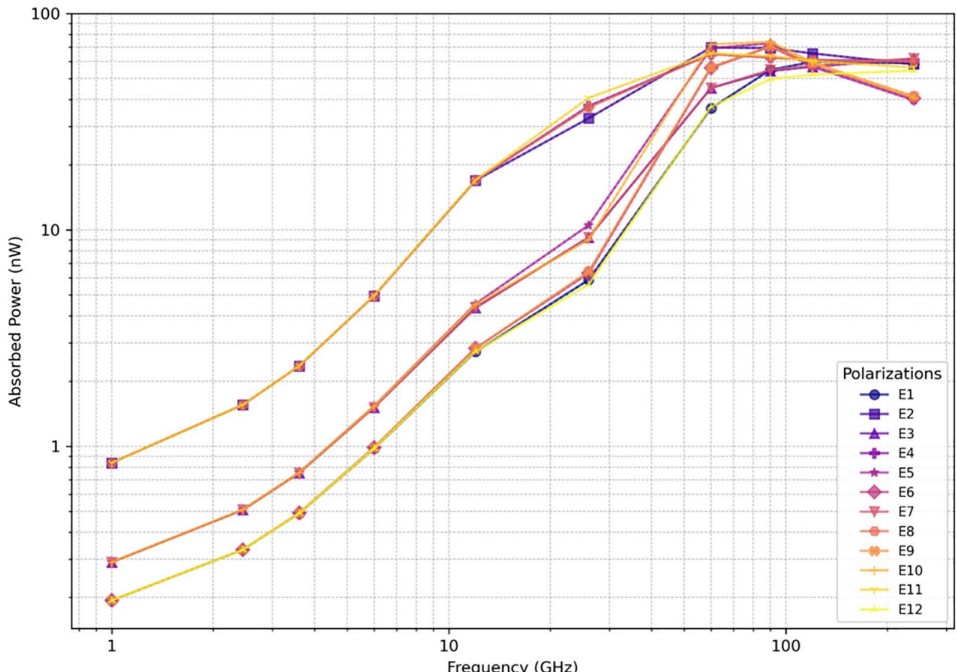

**Fig 4. Total absorbed power (W) in D. melanogaster model from far-field exposure of strength 1 V/m in the frequency range 1-240 GHz in 12 polarizations.** The markers show the results for the individual plane waves.

model of Herssens et al. [12], predicts absorbed power of $1.9 \cdot 10^{-10}$ W, $8.1 \cdot 10^{-10}$ W, $4.2 \cdot 10^{-9}$ W at 6, 12, and 60 GHz, respectively. These values are in excellent agreement with what we found in this study using full wave simulations.

Tissue-specific and whole-body averaged absorbed power for a set of far-field incident plane waves at 3.6 GHz can be found in S4 Table in the Supporting Information. At this frequency, most of the power is absorbed in the internal structures of the fly. Internal organs, particularly the brain, absorb a low amount of power at these frequencies because of their small volume and internal location. Jeladze et al. [44] also found that for a multi-tissue honeybee model, the inner tissues absorbed most of the power at 3.7 GHz. The accuracy of the tissue-specific values can be improved by using tissue-specific dielectric properties, which currently do not exist for this insect. However, the values demonstrate the model's usability in calculating such values.

**Absorbed power in *D. melanogaster* during the behavioral experiments.** On average, the absorbed power in *D. melanogaster* tissue during the behavioral experiments was 3.56 to 9.88 nW, assuming a lower bound and upper bound electric field strength of 5.4 and 9 V/m, respectively. The exact range of dosimetric values for every polarization during the behavioral experiments are listed in S5 Table in the Supporting Information. In our simulations, we did not consider standing waves that could occur due to interference between reflections within the incubator. However, we do take variation in absorption into account by considering a range of potential angles of incidence. We also did not model the presence of the food, the DAM, and the tubes, and only considered plane wave exposure of the fly model in free space.

**Absorbed power in *D. melanogaster* during the fecundity experiments.** The mean absorbed power in *D. melanogaster* tissue during the fecundity experiments for 12 locations and 2 orientations with respect to a 3.6 GHz dipole antenna was simulated to be $1.91 \pm 2.89$ mW. The specific values for every location and orientation are listed in S6 Table. As opposed to the far field simulations, the absorbed power is systematically higher for orthogonal orientations with respect to the antenna arm. There is significant variation in the dose due to the non-uniform nature of the near field. The absorption at the locations close (11 mm) to the antenna (3,7,11) of 0.881–12.3 mW is higher than the one at the opposite side of the tube (locations 4,8,12) at 34 mm from the antenna, 0.215–1.02 mW for an emitted power of 23.5 dBm at 3,6 GHz. Separation distance from the antenna is thus an important factor in this variation. The absorbed dose exceeds the

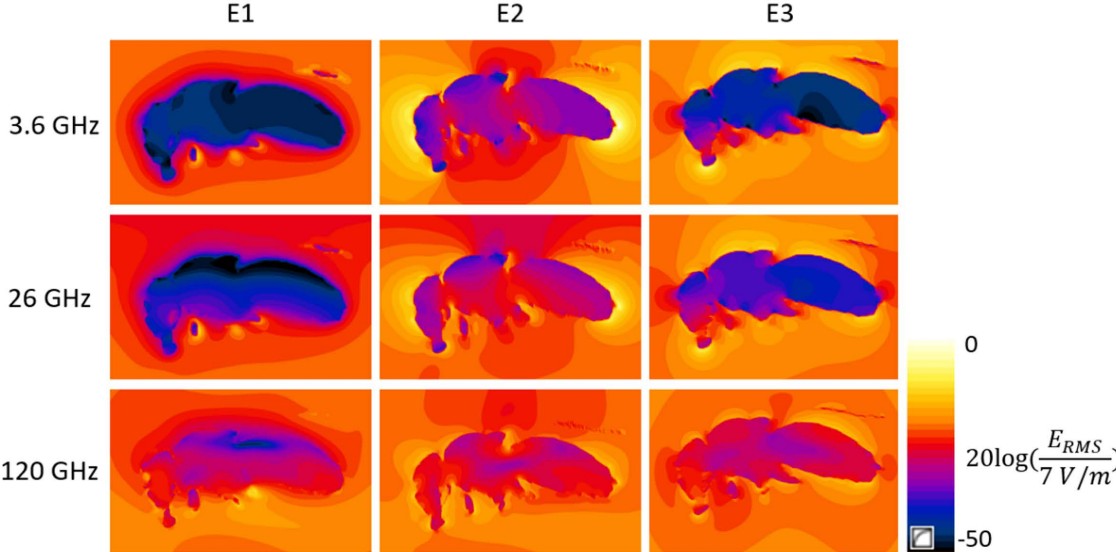

**Fig 5. Median plane cross-sections of the electric field strength (V/m) in and around the D. melanogaster model during far-field exposure of incident field strength 1 V/m at 3.6, 26 and 120 GHz for 3 orthogonal polarizations E1, E2 and E3 of the electric field.**

dose from far field exposure in the behavioral experiments by at least a factor 300. The average absorbed power of 1.91 mW in the near field corresponds to a (polarization-averaged) far-field exposure with an incident field strength of 125 V/m. At 3.6 GHz ICNIRP guidelines prescribe a limit of 61.5 V/m averaged over 30 minutes for the general (human) public [45]. Whereas the far field strength in the behavioral experiments is well below this limit, the equivalent field that would be necessary to induce the same dose as experienced in the near field of the antenna used in the fecundity experiments far exceeds it. Moreover, the variability of the near field can create local hotspots within the insect, which could be more biologically significant than a lower-level, widely distributed absorption.

## Behavioural experiments

The locomotor behaviour of male and female flies was analyzed before, during and after 5G cycles or sham treatment. As expected, flies in all studied groups showed circadian periodicity throughout the experimental interval in constant dark conditions. They did so in each case with an average intrinsic period length of 22.5h, which was then used to calculate activity counts per circadian cycle, plot actograms and circadian activity profiles as well as determine relative rhythmic power. While double-plotted actograms and activity profiles did not reveal obvious differences as a result of 5G versus sham treatment, they did indicate sex differences consistent with prior studies [35] with males exhibiting increased bimodality in their circadian activity patterns (see Fig 6 A, B). Repeated Measures ANOVA and Kruskal-Wallis tests of the impact of treatment interval on relative rhythmic power or period length failed to detect a significant association with one exception (Kruskal-Wallis test $p = 0.04$ for period length of experimental males). Moreover, pairwise comparisons of relative rhythmic power or period length of different intervals for the same flies invariably failed to detect significant differences (see Figs 6 C, D). In contrast, circadian activity counts were found to significantly vary over the three 5-day intervals for three out of the four sex-treatment combinations (see Fig 6E). This was, however, apparently not associated with the consequences of RF exposure, but rather with an increase in activity during the latter two 5-day intervals compared to the first (pre-5G or pre-sham exposure) interval. Separate ANOVA/Kruskal-Wallis analyses of average circadian activity, circadian period and relative rhythmic power were conducted for data sets for each sex that included both the experimental and control time course. Significant associations were detected for treatment interval with average circadian activity counts in females ($p = 0.0006$) and with circadian period length in males ($p = 0.0281$). However, post-hoc tests failed to detect any significant pairwise differences between the pre-5G versus pre-sham, 5G versus sham or post-5G versus post-sham interval comparisons. Taken together, none of our data analyses provide meaningful evidence for an effect of an RF exposure at 3.6 GHz on locomotor activity or circadian clock function. Detailed statistical analysis can be found in Table 1.

Table 1 also lists the comparative testing of the sham-exposed versus RF-exposed flies, separated for male and female flies. We found significant differences in activity counts for female flies and in the periodicity for male flies using the Kruskal-Wallis test at $p < 0.05$ for the entire pre-sham-post sequence versus the entire pre-5G-post sequence. Post-hoc testing of the paired pre-treatment, the sham-exposure versus the RF-exposure, and the post-treatment data showed no significant differences.

Wang et al. [46] also investigated *D. melanogaster* locomotion for 3 days under RF-EMF exposure at 3.5 GHz at 6, 19, and 61 V/m. They found increased activity for male flies at the two higher levels of exposure in comparison to control, but not at the lowest level of exposure. Our exposure levels of between 5.6 and 9 V/m correspond to their lowest exposure level. We also did not find differences in activity between exposed and control male flies (we found an increase for both). An interesting next step in our research would thus be to repeat the assay under higher levels of exposure. We controlled the environmental temperature but did not monitor the temperature of the individual flies in our experiment, as it would have been nearly impossible to do so. Tomioka et al. [47] demonstrated that temperature influences the circadian locomotor rhythm of *D. melanogaster*. Since we did not observe any significant changes in locomotor activity due to RF exposure, we hypothesize that there was no significant heating.

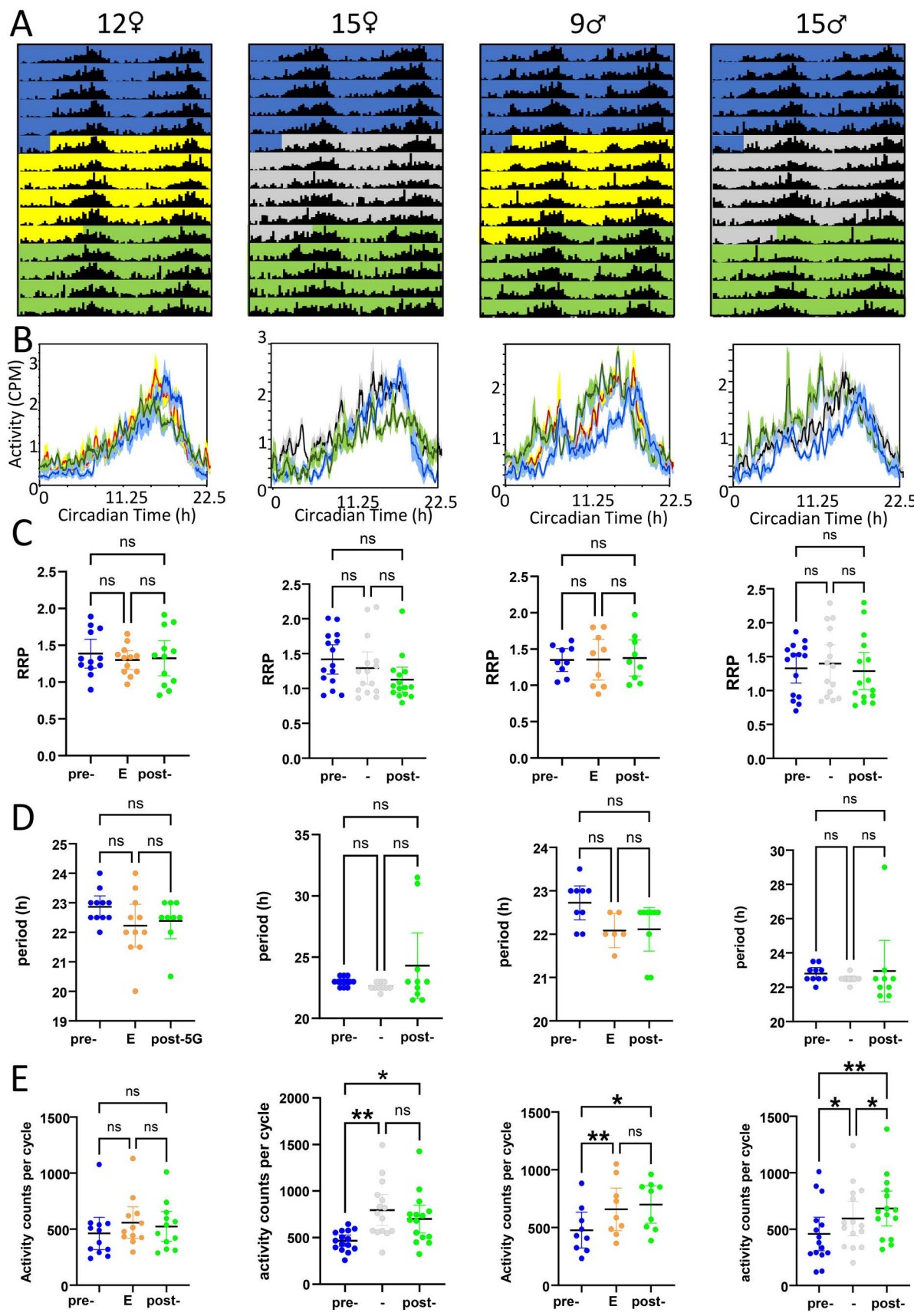

**Fig 6. Impact of 3.6 GHz RF-EMF exposure on locomotor activity rhythms.** Male and female wild-type Berlin-K Drosophila were monitored for their locomotor activity in constant dark conditions at 20°C over the course of 15 days. During the second 5-day interval, experimental flies were exposed to cycles of 12h on/12h off 5G irradiation, while control flies remained unexposed. Each panel represents from left to right: 12 experimental females, 15 control females, 12 experimental males, and 15 control males, respectively. A) Double-plotted modulo-22.5h actograms at 30 min resolution with the first and third 5-day intervals indicated in blue and green, and the 2nd 5-day interval indicated in yellow with orange shading indicating Exposure times and grey for Control (C) conditions. B) Circadian activity profiles at 30-min resolution indicating the activity counts per minute for each of the 5-day intervals. C) Relative rhythmic power (RRP) of locomotor activity by treatment interval. D) Circadian period length for detected rhythms by treatment interval. E) Activity counts per 22.5h cycle by treatment interval.

**Table 1. Statistical analysis for behavioural, locomotor activity experiments** containing Shapiro-Wilk normality test (ns = not significant, * indicates varying degrees of deviation from normality), Kruskal-Wallis (KW) test, RMA (Repeated Measures ANOVA), and ANOVA. $p < 0.05$ was considered statistically significant.

| Format | Sex | Measure | Shapiro-Wilk test | | | Test | Statistic | p | Post-hoc | | | Post-hoc | | |
|---|---|---|---|---|---|---|---|---|---|---|---|---|---|---|
| | | | pre | sham/5G | post | | | | pre-sham/5G | sham/5G-post | pre-post | pre[sham]-pre[5G] | sham-5G | post[sham]-post[5G] |
| pre-sham-post | f | period | * | * | *** | KW | 3.784 | 0.1508 | NA | NA | NA | NA | | |
| pre-sham-post | f | RRP | ns | * | ** | KW | 5.081 | 0.0788 | NA | NA | NA | | | |
| pre-sham-post | f | counts | ns | ns | ns | RMA | 10.88 | **0.0004** | **0.0003** | 0.3995 | **0.0088** | | | |
| pre-sham-post | m | period | ns | *** | **** | KW | 3.823 | 0.1478 | NA | NA | NA | | | |
| pre-sham-post | m | RRP | ns | **ns** | * | KW | 0.4058 | 0.8164 | NA | NA | NA | | | |
| pre-sham-post | m | counts | ns | ns | ns | RMA | 14.33 | **0.0003** | **0.0176** | **0.0252** | **0.0017** | | | |
| pre-5G-post | f | period | ns | ns | ** | KW | 3.579 | 0.1671 | NA | NA | NA | | | |
| pre-5G-post | f | RRP | ns | ns | ns | RMA | 0.5035 | 0.5957 | NA | NA | NA | | | |
| pre-5G-post | f | counts | ** | * | ns | KW | 2.168 | 0.3382 | NA | NA | NA | | | |
| pre-5G-post | m | period | ns | ns | *** | KW | 6.471 | **0.0393** | 0.0581 | >0.9999 | 0.1567 | | | |
| pre-5G-post | m | RRP | ns | ns | ns | RMA | 0.04376 | 0.9524 | NA | NA | NA | | | |
| pre-5G-post | m | counts | ns | ns | ns | RMA | 9.262 | **0.0059** | **0.0088** | 0.6729 | **0.0295** | | | |
| pre-sham-post vs pre-5G-post | f | period | see above | | | KW | 9.149 | 0.1033 | see above | | | NA | NA | NA |
| pre-sham-post vs pre-5G-post | f | RRP | | | | KW | 7.948 | 0.1591 | | | | NA | NA | NA |
| pre-sham-post vs pre-5G-post | f | counts | | | | KW | 21.53 | **0.0006** | | | | >0.9999 | 0.0591 | 0.1886 |
| pre-sham-post vs pre-5G-post | m | period | | | | KW | 12.54 | **0.0281** | | | | >0.9999 | 0.2706 | >0.9999 |
| pre-sham-post vs pre-5G-post | m | RRP | | | | KW | 1.067 | 0.957 | | | | NA | NA | NA |
| pre-sham-post vs pre-5G-post | m | counts | | | | A | 2.009 | 0.0887 | | | | NA | NA | NA |

## Fecundity experiments

Fig 7 shows the results of the fecundity experiments during various time intervals. The RF EMF exposure (and control conditions) happened during days 1 and 2. Two-way ANOVA analysis found a significant effect of time interval on fecundity ($F_{(4,50)} = 18.38$, $p < 0.0001$), but no impact of treatment ($F_{(1,50)} = 0.0009592$, $p = 0.9754$) or the time interval x treatment interaction ($F_{(4,50)} = 1.554$, $p = 0.2012$). Detailed statistical analysis can be found in S8 and S9 Tables. Post-hoc pairwise comparisons of different time intervals found significantly lower fecundity during day 1 and 2 versus all other days as well as significantly lower fecundity during day 3 versus day 5 or 6. Thus, fecundity increased from lower levels during the early days of the experiment, regardless of treatment. This time-dependent increase was expected based on prior

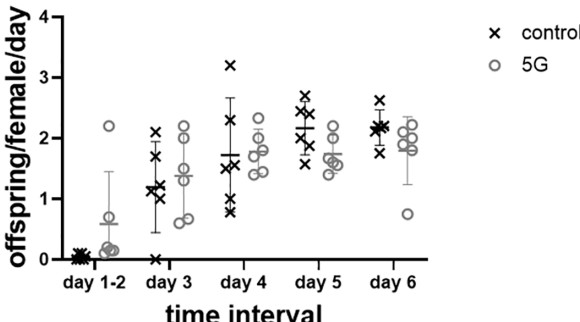

**Fig 7. Number of offspring per female per day for an exposed (5G) group alongside the control group.**

studies of Drosophila fecundity over time [48,49]. We can conclude that at this exposure level at 3.6 GHz, no effect on fecundity was observed.

Prior studies on this topic have shown mixed results. Pay et al. [4] heated female *D. melanogaster* using 2.45 GHz RF-EMFs and found a reduction in fecundity in comparison to sham exposure. They observed the same effect using an alternative heating method. This confirmed that, as for any other species, also for *D. melanogaster* there is a thermal level of RF-EMF exposure that will induce adverse effects. However, it is as yet unclear where exactly the threshold for this effect would be. Dardalhon et al. [50] found an increase in fertility for female wild type *D. melanogaster* exposed to RF-EMFs at 17 GHz for 16-21h at 60 mW/cm² or 476 V/m, when they mated with untreated males. The absorbed power at this higher frequency and higher field strength must have been significantly higher than in our study, but presumably lower than in [4]. Alti and Ünlü [51] found a reduction in offspring for females exposed to 10 GHz at 3.42 V/m during maximally 6 hours in the larval stage. While the exposure level was similar to the one we used, exposure during a different life stage makes it difficult to compare. The absence of an effect in our experiments, contrasted with the findings of [51] at similar exposure levels, suggests that *D. melanogaster* may be more susceptible to RF-EMF–induced effects during development than in adulthood. A future study on this topic might thus look into RF-EMF exposure of the larval or pupal phase. Panagopoulos et al. [18,19] found a reduction in offspring for female *D. melanogaster* exposed to a mobile phone presumably emitting RF-EMFs at 900 and 1800 MHz, which were reported to be around 14 V/m in the vials close to the emitting phone, but are more likely higher given the GSM technology used in that paper with output powers up to 1 W. Interestingly, Weisbrot et al. [20] found exactly the opposite effect, an increase in offspring, for the same experiment. While we controlled the environmental temperature, we did not monitor the temperature of the individual flies during the RF exposure. Hence we cannot exclude the possibility of heating occurring. However, Ganadara et al [52] and Huy et al. [53] demonstrated that temperature has an influence on D melanogaster fertility. Since we did not observe any effect, we hypothesize that there was no significant heating in our experiments. Given the above, we conclude that more research is needed to conclusively determine where the thresholds are for RF-EMF absorption, subsequent dielectric heating, and effects on fecundity in insects.

In the fecundity and behavioral experiments, we used continuous RF-EMF exposure, while real telecommunication networks use modulated signals to transmit information. In our follow-up studies, we will use modulated signals to fully emulate RF-EMF exposure in telecommunication networks.

## Conclusion

This paper investigated RF dosimetry of adult *D. melanogaster* and used this dosimetry to investigate whether certain levels of 3.6 GHz RF-EMF absorption in an adult fruit fly are associated with behavioural or fecundity effects. We performed micro-CT scans on a female D. melanogaster specimen and used these scans to create a multi-tissue 3D digital model of the fly. We were

able to use this in numerical EM simulations to estimate the absorbed power in a fly as a function of RF frequency. We found that under constant far-field RF-EMF exposure adult *D. melanogaster* show a peak in RF absorption around 90 GHz, where an exposure to 1 V/m corresponds to an absorption of 6 nW. Below this frequency, the absorption increases as a function of frequency.

In parallel, we ran two assays looking into potential effects on the locomotor activity and fecundity of *D. melanogaster* exposed to 3.6 GHz. In the behavioural experiments no effects were found on the locomotor activity for a 5-day exposure to RF field values between 5.4 and 9 V/m, which correspond to 3.56 nW to 9.88 nW absorbed power. A next step here would be to repeat this assay under higher levels of exposure. In terms of fecundity, we found that an absorption level at 3.6 GHz of on average 1.91 mW during 48h did not affect fecundity of adult female flies. Our next plan in this direction would be to investigate RF-EMF exposure of the larval or pupal phase rather than the adult stages.

## Supporting information

**S1 File. Extended Methods.**
(DOCX)

**S2 File. Extended Results.**
(DOCX)

**S1 Table. Dielectric properties of insect tissues at 1–240 GHz as obtained in [Thielens et al., 2018].**
(DOCX)

**S2 Table. Measured RF-EMF Electric Field strengths (in V/m) corresponding to the behavioural experiments. Columns indicate different conditions and measurement devices, while the rows indicate different measurement locations.**
(DOCX)

**S3 Table. Measured RF-EMF power corresponding to the fecundity experiments. Columns indicate different conditions, while the rows indicate different measurement locations. The measurements were carried out with the RF Explorer (Seeed Studio) in N = 30 repetitions.**
(DOCX)

**S4 Table. Absorbed power (nW) in D. melanogaster tissues from 1 V/m far-field exposure at 3.6 GHz for 12 polarizations (E1 to E12).**
(DOCX)

**S5 Table. Absorbed power (nW) in D. melanogaster tissue during the behavioral experiments under the assumption of a lower bound and upper bound electric field strength of 5.4 and 9 V/m resp. These were obtained by rescaling the total absorbed power that was computed in the simulations by the squared electric field strength in the behavioural experiments.**
(DOCX)

**S6 Table. Absorbed power (mW) in D. melanogaster tissue during the fecundity experiments for 12 locations and 2 orientations w.r.t. a 3.6 GHz dipole antenna with an input power of 23.5 dBm.**
(DOCX)

**S7 Table. ANOVA test showing how treatment interaction, treatment type and time interval contribute to the variation in the fecundity experiments, where SS = Sum of Squares, DF = Degrees of Freedom, MS = Mean Square, F = F-statistic. p < 0.05 was considered statistically significant.**
(DOCX)

**S8 Table. Tukey's multiple comparisons test as post-hoc analysis following significant variation with time interval found in ANOVA test. p<0.05 was considered statistically significant.**
(DOCX)

**S1 Fig. Simulation configuration of the fecundity experiments showing 12 locations of the Drosophila model each with two orientations (orthogonal ⊥ and parallel ∥ w.r.t. the dipole antenna on the left).**
(DOCX)

## Acknowledgments

We acknowledge the µ-VIS X-ray Imaging Centre (muvis.org), part of the National Facility for laboratory-based X-ray CT (nxct.ac.uk – EPSRC: EP/T02593X/1) at the University of Southampton for the provision of the µCT imaging infrastructure as well as the Insectary of the invertebrate Facility of the School of Biological Sciences of the University of Southampton for support of the behavioral studies.

## Author contributions

**Conceptualization:** Pieterjan De Boose, Duncan Bell, Jürg Fröhlich, Anke Huss, Orestis L. Katsamenis, Lewis Blackwell, Pantelitsa Georgiadou, Ayesha Mohiud-din, Elham Alshamrani, Christelle Lasbleiz, Herman Wijnen, Arno Thielens.

**Data curation:** Pieterjan De Boose, Felipe Oliveira Ribas, Orestis L. Katsamenis, Lewis Blackwell, Pantelitsa Georgiadou, Ayesha Mohiud-din, Elham Alshamrani, Herman Wijnen.

**Formal analysis:** Pieterjan De Boose, Orestis L. Katsamenis, Pantelitsa Georgiadou, Ayesha Mohiud-din, Elham Alshamrani, Herman Wijnen, Arno Thielens.

**Funding acquisition:** Maria Bouga, Jürg Fröhlich, Fani Hatjina, Anke Huss, Menelaos Stavrinides, Antonios Tsagkarakis, Marco Zahner, Herman Wijnen, Arno Thielens.

**Investigation:** Pieterjan De Boose, Duncan Bell, Eline De Borre, Orestis L. Katsamenis, Lewis Blackwell, Pantelitsa Georgiadou, Ayesha Mohiud-din, Elham Alshamrani, Herman Wijnen, Arno Thielens.

**Methodology:** Pieterjan De Boose, Felipe Oliveira Ribas, Duncan Bell, Jürg Fröhlich, Anke Huss, Anastasia Kalapouti, Orestis L. Katsamenis, Lewis Blackwell, Pantelitsa Georgiadou, Ayesha Mohiud-din, Elham Alshamrani, Christelle Lasbleiz, Zoi Thanou, Antonios Tsagkarakis, Andri Varnava, Marco Zahner, Herman Wijnen, Arno Thielens.

**Project administration:** Maria Bouga, Anke Huss, Christelle Lasbleiz, Menelaos Stavrinides, Antonios Tsagkarakis, Herman Wijnen, Arno Thielens.

**Resources:** Orestis L. Katsamenis, Herman Wijnen, Arno Thielens.

**Software:** Felipe Oliveira Ribas, Orestis L. Katsamenis, Herman Wijnen.

**Supervision:** Herman Wijnen, Arno Thielens.

**Validation:** Pieterjan De Boose.

**Visualization:** Pieterjan De Boose, Felipe Oliveira Ribas, Eline De Borre, Orestis L. Katsamenis, Herman Wijnen, Arno Thielens.

**Writing – original draft:** Pieterjan De Boose, Orestis L. Katsamenis, Herman Wijnen, Arno Thielens.

**Writing – review & editing:** Pieterjan De Boose, Felipe Oliveira Ribas, Duncan Bell, Maria Bouga, Eline De Borre, Jürg Fröhlich, Fani Hatjina, Anke Huss, Anastasia Kalapouti, Orestis L. Katsamenis, Lewis Blackwell, Christelle Lasbleiz, Menelaos Stavrinides, Zoi Thanou, Antonios Tsagkarakis, Andri Varnava, Marco Zahner, Herman Wijnen, Arno Thielens.

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
