## [Decision Letter · Decision Letter 0]

18 Aug 2025

Dear Dr. Arno Thielens,

Thank you for submitting your manuscript to PLOS ONE. After careful consideration, we feel that it has merit but does not fully meet PLOS ONE’s publication criteria as it currently stands. Therefore, we invite you to submit a revised version of the manuscript that addresses the points raised during the review process

We look forward to receiving your revised manuscript.

Kind regards,

Olga Zeni

Academic Editor

PLOS ONE

“European Union’s Horizon Europe research and innovation funding program under grant agreement No 101057216 (ETAIN).”

“Part of this work is funded by the European Union’s Horizon Europe research and innovation funding program under grant agreement No 101057216 (ETAIN). We also acknowledge the μ-VIS X-ray Imaging Centre (muvis.org), part of the National Facility for laboratory-based X-ray CT (nxct.ac.uk – EPSRC: EP/T02593X/1) at the University of Southampton for the provision of the μCT imaging infrastructure as well as the Insectary of the invertebrate Facility of the School of Biological Sciences of the University of Southampton for support of the behavioral studies.”

“European Union’s Horizon Europe research and innovation funding program under grant agreement No 101057216 (ETAIN).”

Additional Editor Comments:

The reviewers raised important criticalities in the methodology that need to be carefully addressed to make the manuscript acceptable for publication.

Reviewers' comments:

Reviewer's Responses to Questions

**Comments to the Author**

1. Is the manuscript technically sound, and do the data support the conclusions?

Reviewer #1: Partly

Reviewer #2: Partly

2. Has the statistical analysis been performed appropriately and rigorously?

Reviewer #1: Yes

Reviewer #2: N/A

3. Have the authors made all data underlying the findings in their manuscript fully available?

Reviewer #1: Yes

Reviewer #2: Yes

4. Is the manuscript presented in an intelligible fashion and written in standard English?

Reviewer #1: Yes

Reviewer #2: Yes

Reviewer #1: The study analyses the impact of Drosophila melanogaster exposure to 3.6 GHz. The topic is very important given the insects' constant exposure to radiofrequency (RF) electromagnetic fields and their fundamental role in the ecosystem. The authors created a 3D model of the animal, used for numerical dosimetry, then investigated two important biological endpoints following RF exposure. The authors find no effect on locomotor activity and fecundity.

The experiments are clearly described however the conclusions are weakened by the lack of some controls in the experimental setup. The main critical issues are listed below.

Major

1) Lack of experimental dosimetry.

2) In the absence of the positive control, the validity of the methods (behavioural and fecundity experiments) cannot be confirmed.

3) For behavioural experiments, lack of negative control (I mean samples hosted in a standard environment/incubator to provide info on the background level of the endpoint under examination) to ensure that the exposure system does not affect the endpoint analysed.

4) For fecundity experiments, the sham described seems to be a negative control (see above) rather than a sham. In addition, it seems that temperature is controlled but not the humidity of the room.

5) The authors should clarify whether the exposure conditions cause heating of the sample or not.

6) The authors should state whether the experiments are conducted blindly.

Minor

For behavioural experiments

1) The choice of exposure schedule (5 days before, 5 days of exposure, 5 days after) should be justified.

2) The authors claim to use 32 flies for the experiments (line 11 of page 8) but in Figure 6, 9 exposed males are shown, and in the caption of the same figure, 12 exposed males are mentioned. Please clarify this inconsistency.

3) The aim of the paper is to analyse the possible effect of RF exposure so in my opinion the comparison of interest, to be reported in the main text rather than in the supplementary, is RF vs sham and not those reported in Figure 6 (pre RF vs RF, RF vs post RF and pre RF vs post RF).

For fecundity experiments

4) Please provide rationale for the duration of the exposure (2 days).

5) Clarify whether the exposure is continuous or intermittent.

Reviewer #2: Though some aspects of the work is well described, there is a lack of important detail when considering the experimental setup and the dosimetry, to the extent that the experiments could not be replicated and some values provided are meaningless without further information. The paper could be improved by the addition of the missing information.

It should be clearly stated that exposures are to continuous wave and as such they do not represent the exposures that insects in the environment of a 5G mobile telecommunications installation would be exposed to which are characterized by high peak to average power ratios and TDD on-off slots for up and down link.

Fecundity Experiments

In this case the experiment geometry was well described, however, it would appear that the food which occupied a significant volume was not modelled nor were the dielectric properties provided. How was the exposure influenced when the flies were close to the food, how did this impact the uncertainty? The diameter of the tube was significant when compared to the spacing of the tube to the dipole antenna, the radial variation in the exposure field should be noted along with the axial variation – though it does seem that this was taken into account in the reported absorbed power levels, the link of the variation to that of the exposure field would be a nice addition. Please add information on the light dark cycles or state use for these experiments. The use of the Narda NBM (the probe type is not mentioned and should be) to measure in the near field is not appropriate due to the large averaging volume of the sensors in what is probably a 66mm diameter probe.

In the supplementary information it is stated that at 30cm from the dipole the power was 1.32nW in simulation and 1.29nW measured by the Seeed RF Explorer, without the antenna factor for this probe the results are rather meaningless, please convert to power densities i.e. W/m2, presumably this can be done as you must have a model to have presented a simulated value.

On page 12 you talk about the volatility of the near field, do you mean variability?

Behavioral Experiments

This experiment could not be replicated as there are key details of the experiment that have not been supplied. The spacing of the antenna from the metallic grid is not stated, as this antenna radiates in both directions this is essential information. It is not made clear to the reader that the exposure is in a standing wave due to reflection from the bottom of the incubator, so this dimension is also essential to know. The standing wave is however taken into account in the extremes of the exposure over the whole exposure volume. Is there any influence on the exposure field by scattering due to the regular array of food in the exposure volume, what is its impact on the exposure uncertainty, the dielectric properties of the tube support (and diensions) and the food is not provided.

Numerical dosimetry

Though the model generated from the micro CT is very detailed with various anatomical structures identified, the dosimetry uses a model with all tissues set to the same dielectric parameters, i.e. homogeneous, these parameters obtained from homogenized whole insects. What is the density of the tissue, this can then be used to determine measures such as average SAR. Therefore, the information in table S4 must be at the very least extremely speculative and with very high uncertainty, is its inclusion justified.

**Do you want your identity to be public for this peer review?** For information about this choice, including consent withdrawal, please see our Privacy Policy

Reviewer #1: **Yes: ** Mariateresa Allocca

Reviewer #2: No

---

## [Author Response · Author response to Decision Letter 1]

2 Oct 2025

We have addressed the reviewer's comments in a separate file, which is uploaded alongside the manuscript.

---

## [Decision Letter · Decision Letter 1]

22 Oct 2025

Behaviour and Reproduction of Drosophila Melanogaster Exposed to 3.6 GHz Radio-Frequency Electromagnetic Fields

PONE-D-25-18660R1

Dear Dr. Arno Thielens,

We’re pleased to inform you that your manuscript has been judged scientifically suitable for publication and will be formally accepted for publication once it meets all outstanding technical requirements.

Kind regards,

Olga Zeni

Academic Editor

PLOS ONE

Additional Editor Comments (optional):

Major issues have been adressed in the revised manuscript, although improvement on dosimetry and experimental controls should be considered in future studies.

Reviewers' comments:

Reviewer's Responses to Questions

**Comments to the Author**

Reviewer #1: All comments have been addressed

Reviewer #2: All comments have been addressed

2. Is the manuscript technically sound, and do the data support the conclusions?

Reviewer #1: Yes

Reviewer #2: Yes

3. Has the statistical analysis been performed appropriately and rigorously?

Reviewer #1: Yes

Reviewer #2: I Don't Know

4. Have the authors made all data underlying the findings in their manuscript fully available?

Reviewer #1: Yes

Reviewer #2: Yes

5. Is the manuscript presented in an intelligible fashion and written in standard English?

Reviewer #1: Yes

Reviewer #2: Yes

Reviewer #1: I would like to thank the authors for responding to my requests wherever possible. I understand that some experimental constraints arise from in vivo experimentation, and I hope that greater efforts will be made in providing appropriate controls to improve the quality of these studies.

Reviewer #2: The paper is very much improved and all comments addressed.

It would be great if for future work exposure conditions that are more homogeneous be employed and better dosimetry be performed. Ideally modulated signals should be used particularly for behavioral experiments

**Do you want your identity to be public for this peer review?** For information about this choice, including consent withdrawal, please see our Privacy Policy

Reviewer #1: **Yes: ** Mariateresa Allocca

Reviewer #2: No

---

## [Editor Report · Acceptance letter]

PONE-D-25-18660R1

PLOS ONE

Dear Dr. Thielens,

I'm pleased to inform you that your manuscript has been deemed suitable for publication in PLOS ONE. Congratulations! Your manuscript is now being handed over to our production team.

Kind regards,

on behalf of

Dr. Olga Zeni

Academic Editor

PLOS ONE